| Editor's Pick | Computational Biology | Resource Report

# Rocket-miR, a translational launchpad for miRNA-based antimicrobial drug development

Samuel L. Neff,[1] Thomas H. Hampton,[1] Katja Koeppen,[1] Sharanya Sarkar,[1] Casey J. Latario,[1] Benjamin D. Ross,[1] Bruce A. Stanton[1]

**ABSTRACT** Developing software tools that leverage biological data sets to accelerate drug discovery is an important aspect of bioinformatic research. Here, we present a novel example: a web application called Rocket-miR that applies an existing bioinformatic algorithm (IntaRNA) to predict cross-species miRNA-mRNA interactions and identify human miRNAs with potential antimicrobial activity against antibiotic-resistant bacterial infections. Rocket-miR is the logical extension of our prior finding that human miRNA let-7b-5p impairs the ability of the ubiquitous opportunistic pathogen *Pseudomonas aeruginosa* to form biofilms and resist the bactericidal effect of β-lactam antibiotics. Rocket-miR's point and click interface enables researchers without programming expertise to predict additional human-miRNA-pathogen interactions. Identified miRNAs can be developed into novel antimicrobials effective against the 24 clinically relevant pathogens, implicated in diseases of the lung, gut, and other organs, that are included in the application. The paper incorporates three case studies contributed by microbiologists that study human pathogens to demonstrate the usefulness and usability of the application. Rocket-miR is accessible at the following link: http://scangeo.dartmouth.edu/RocketmiR/.

**IMPORTANCE** Antimicrobial-resistant infections contribute to millions of deaths worldwide every year. In particular, the group of bacteria collectively known as ESKAPE (*Enterococcus faecium, Staphylococcus aureus, Klebsiella pneumoniae, Acinetobacter baumannii, Pseudomonas aeruginosa,* and *Enterobacter* sp.) pathogens are of considerable medical concern due to their virulence and exceptional ability to develop antibiotic resistance. New kinds of antimicrobial therapies are urgently needed to treat patients for whom existing antibiotics are ineffective. The Rocket-miR application predicts targets of human miRNAs in bacterial and fungal pathogens, rapidly identifying candidate miRNA-based antimicrobials. The application's target audience are microbiologists that have the laboratory resources to test the application's predictions. The Rocket-miR application currently supports 24 recognized human pathogens that are relevant to numerous diseases including cystic fibrosis, chronic obstructive pulmonary disease (COPD), urinary tract infections, and pneumonia. Furthermore, the application code was designed to be easily extendible to other human pathogens that commonly cause hospital-acquired infections.

**KEYWORDS** cystic fibrosis, bioinformatics, miRNA, CF pathogens, antimicrobial agents, host-pathogen interactions, ESKAPE pathogens

Antimicrobial resistance (AMR) is a chief global health threat of the 21st century. AMR infections affect more than 2.8 million people and cause 35,000 deaths in the USA per year according to a recent report by the Centers for Disease Control and Prevention (CDC) (1). A systematic literature review in *The Lancet*, drawing from hospital records and public health surveillance data, attributed over 1.2 million deaths worldwide to bacterial

Address correspondence to Bruce A. Stanton, bruce.a.stanton@dartmouth.edu.

The authors declare no conflict of interest.

See the funding table on p. 20.

AMR in 2019, the majority of which were caused by lower respiratory tract infections (2). The Organization for Economic Co-operation and Development further estimated in 2018 that over the next 30 years (2020–2050), managing the complications of AMR will carry a cost of $3.5 billion per year to the global economy (3). Unfortunately, for a variety of technical and economic reasons, relatively few drug and biotechnology companies are developing new antibiotics (4). Thus, there is a pressing need for novel therapies to treat AMR.

The risk of AMR is especially acute for people with chronic diseases, such as cystic fibrosis (CF) and chronic obstructive pulmonary disease (COPD), that render individuals immune-compromised and prone to chronic antibiotic-resistant infections. The application described in this paper was specifically developed to identify human miRNAs that target AMR genes in CF-related pathogens but is relevant to drug-resistant pathogens in general. Common CF pathogens *Pseudomonas aeruginosa* and *Staphylococcus aureus*, for example, are also listed among the six ESKAPE (*Enterococcus faecium, Staphylococcus aureus, Klebsiella pneumoniae, Acinetobacter baumannii, Pseudomonas aeruginosa,* and *Enterobacter* sp.) pathogens, a group of antibiotic-resistant bacteria that are a common cause of hospital-acquired infections and recognized by the WHO as pathogens in urgent need of new antibiotics (5, 6).

*Pseudomonas aeruginosa* is the most prevalent lung pathogen in people with CF (pwCF), especially adults, and a significant contributor to morbidity and mortality (7, 8). In the CF lungs, mutations in the cystic fibrosis transmembrane conductance regulator (*CFTR*) gene lead to the production of highly viscous, hyper-concentrated mucus (9, 10). The physiological consequences are obstructed breathing, dysregulation of the immune system, and poor airway clearance that enables infection by opportunistic pathogens, notably drug-resistant *P. aeruginosa* (11, 12). A recent study found that the risk of death for pwCF doubled with *P. aeruginosa* lung infection and increased eightfold when the bacteria acquired resistance to antibiotics (13). Furthermore, a recent meta-analysis involving *P. aeruginosa* isolates treated with common anti-pseudomonal antibiotics determined that for a majority of the 22 antibiotics surveyed, over one-third of clinical isolates are resistant (14). A range of other pathogens are also known to contribute to CF lung disease, such as *Burkholderia cenocepacia*, *Stenotrophomonas maltophilia*, *S. aureus* (particularly methicillin-resistant strains), *Mycobacterium abscessus*, and the fungal species *Aspergillus fumigatus* and *Candida albicans* (15, 16). For many of these pathogens, researchers have documented increases in the proportion of bacterial strains resistant to common antibiotics over the past several decades (17–22).

AMR occurs through a variety of well-known mechanisms. In *P. aeruginosa*, for example, AMR is driven by a range of biological changes, including the transition to a biofilm-forming state, increased activity of drug efflux pumps, and the development of β-lactamase activity (23, 24). In response to the failure of conventional antibiotics, researchers are turning to more innovative forms of antimicrobial therapy. Yale University recently launched its Center for Phage Biology and Therapy, with certain researchers focused on developing phage therapy for CF pathogens including *P. aeruginosa* (25–27). Scientists at other institutions are working to develop phage therapy for additional antibiotic-resistant CF pathogens like the non-tuberculosis mycobacteria species, including *M. abscessus*, or the CF pathogen *B. cenocepacia* (28, 29).

Another potential strategy for treating AMR infections is to use miRNAs as antimicrobial therapeutics. Most ongoing efforts at miRNA-based drug development have involved modulating the expression of miRNAs to suppress the genes that contribute to cancer (30–34), though recent studies have involved other diseases including Huntington's and keloid disease (35–37). In addition to the use of human miRNAs to target cancer-causing genes in human cells, we have shown that human miRNA let-7b-5p inhibits several genes that contribute to biofilm formation and antibiotic resistance in *P. aeruginosa* (38).

The potential for cross-kingdom miRNA targeting interactions has been recognized for well over a decade. Studies since the early 2010s have demonstrated that small

RNAs are transferred between crop species and parasitic fungal pathogens—a natural plant defense mechanism that limits fungal growth (39, 40). Recognizing the biological possibility of this plant-fungal exchange, crop scientists have developed genetic engineering techniques to augment the expression of plant small RNAs as a strategy to limit pathogen growth and preserve crop yield (41, 42). Cross-kingdom interactions have also been observed in the human gut microbiome. Researchers have demonstrated the ability of human miRNA mimics to enter and reduce the growth rates of intestinal pathogens (43, 44). Conversely, other studies have utilized human and plant-derived miRNAs to enhance the growth or metabolite production of commensal gut bacteria and ameliorate disease symptoms. Furthermore, it is known that bacteria are capable of releasing small RNAs that influence protein expression in other bacteria (45), human cells (46, 47), or plant cells (48–50).

More recent research has identified cross-kingdom interactions in the lungs, including our own prior work (38, 51, 52). We have previously shown that human miRNA let-7b-5p is delivered through extracellular vesicles (EVs) secreted by human airway cells to *P. aeruginosa*. Once delivered, let-7b-5p decreases the abundance of proteins essential for biofilm formation. Let-7b-5p also reduces resistance to β-lactam antibiotics by inhibiting the expression of antibiotic efflux pumps (38). Moreover, other studies have shown that let-7b-5p is associated with reduced inflammation, suggesting additional beneficial effects, especially in pwCF who have a chronic hyperinflammatory response to chronic infection by *P. aeruginosa* and other CF pathogens (53, 54).

A notable characteristic of miRNAs is that one miRNA may target hundreds of different genes (55–57). Thus, although bacteria can rapidly develop antimicrobial resistance by altering the expression of a few genes, miRNA targeting of hundreds of genes make it more challenging for microbes to develop antibiotic resistance. The research on miRNAs, including our work on let-7b-5p, suggests that it is possible to develop and deliver miRNA-based therapeutics, with or without antibiotics, in engineered extracellular vesicles to eliminate antibiotic-resistant infections. This therapeutic approach could theoretically be applied to all microbial pathogens including bacterial and fungal infections.

The development of Rocket-miR (available online at the following link: http://scangeo.dartmouth.edu/RocketmiR/) described herein enables scientists to target genes and pathways that potentiate antimicrobial resistance by providing miRNA target predictions for 24 clinically relevant pathogens (Table 1). The application leverages the popular bioinformatic tool IntaRNA to make miRNA-targeting predictions (described in detail in Materials and Methods) and provides a user-friendly environment to explore this prediction data and arrive at translational insights. Typically, miRNA target prediction requires extensive computational skill, and months of time to process and analyze prediction data. Our application requires no prior computational experience or effort to process and analyze data. The interactive, visual nature of the Rocket-miR application enables researchers to explore the biological potential of human miRNAs targeting genes and whole genetic pathways in human pathogens in a matter of minutes. Furthermore, genes and pathways in the application are represented by descriptive annotations—gene names, protein functions, and pathway identifiers that are recognizable to biologists. To the best of our knowledge there are no other programs that allow the rapid identification of miRNAs to target antimicrobial-resistant genes in microbial pathogens.

## RESULTS

### Application overview

The Rocket-miR application has five core sections (Fig. 1). The first section is the "Summary View," which offers a high-level perspective on how well the whole set of 630 human miRNAs incorporated in the application target the genes of a selected microbe. The other four sections—the "miRNA View," "Pathway View," "Compare Species" view, and the "Structural Analysis" view—offer a closer look at individual miRNAs, genes, and

**TABLE 1** The 24 clinically relevant human pathogens that can be studied using the Rocket-miR application

| Species | Strain | Type |
| --- | --- | --- |
| *Achromobacter xylosoxidans* | A8 | Gram-negative bacterium |
| *Acinetobacter baumannii* | AYE | Gram-negative bacterium |
| *Aspergillus fumigatus* | Strain 293 | Fungus |
| *Bacteroides fragilis* | NCTC 9343 | Gram-negative bacterium |
| *Burkholderia cenocepacia* | J2315 | Gram-negative bacterium |
| *Burkholderia multivorans* | ATCC 17616 (Tohoku) | Gram-negative bacterium |
| *Candida albicans* | SC5314 | Fungus |
| *Clostridium difficile* | Strain 630 | Gram-positive bacterium |
| *Clostridium perfringens* | ATCC 13124 | Gram-positive bacterium |
| *Enterobacter cloacae* | ATCC 13047 | Gram-negative bacterium |
| *Enterococcus faecium* | DO | Gram-positive bacterium |
| *Escherichia coli* | K12 (Substr. MG1655) | Gram-negative bacterium |
| *Haemophilus influenza* | Strain 477 | Gram-negative bacterium |
| *Klebsiella pneumoniae* | MGH 78578 | Gram-negative bacterium |
| *Mycobacterium abscessus* | ATCC 19977 | Gram-positive bacterium |
| *Mycobacterium avium* | Strain 104 | Gram-positive bacterium |
| *Prevotella melaninogenica* | ATCC 25845 | Gram-negative bacterium |
| *Pseudomonas aeruginosa* | PAO1 | Gram-negative bacterium |
| *Staphylococcus aureus* | NCTC 8325 | Gram-positive bacterium |
| *Stenotrophomonas maltophilia* | K279a | Gram-negative bacterium |
| *Streptococcus parasanguinis* | ATCC 15912 | Gram-positive bacterium |
| *Streptococcus pneumoniae* | R6 | Gram-positive bacterium |
| *Streptococcus salivarius* | CCHSS3 | Gram-positive bacterium |
| *Streptococcus sanguinis* | SK36 | Gram-positive bacterium |

biological pathways. The "miRNA View" allows researchers to identify Kyoto Encyclopedia of Genes and Genomes (KEGG) pathways that are most likely to be effectively targeted by each human miRNA. Users can follow external links to KEGG pathway images highlighting which genes in that pathway met user-defined targeting thresholds. Whether or not a pathway as a whole is predicted to be "effectively targeted" depends on the proportion of its genes/proteins that meet the targeting cutoffs set up in the application. These cutoffs, in turn, are based on prior published experimental data demonstrating the range of energy scores associated with significant downregulation of bacterial protein expression (the targeting cutoffs are described in more detail in Materials and Methods). The "Pathway View" offers the reverse perspective, showing the user a ranked list of human miRNAs that best target a chosen pathway of interest. The "Compare Species" view allows the user to pick multiple microbes and see how well a chosen human miRNA performs across the different species. Finally, the "Structural Analysis" view allows the user to examine and compare the nucleotide sequences of selected miRNAs.

To demonstrate the usefulness of the application in a domain-specific context, Results includes three case studies from researchers who study cystic fibrosis pathogens at the Geisel School of Medicine. Prior to the case studies, we discuss how the vast array of prediction data in the application can be narrowed in scope to identify the miRNAs with the greatest antimicrobial potential.

## Identifying miRNAs with antimicrobial potential

The Rocket-miR application contains a universe of miRNA-targeting predictions which could be validated by laboratories that study AMR pathogens. How are researchers who use the application to decide on the most promising human miRNAs for *in vitro* studies? Surveying and comparing the predicted targets of human miRNAs across all Rocket-miR pathogens illuminate several strategies to identify the most promising miRNAs and species to target. It also helps identify miRNAs with potential broad-spectrum

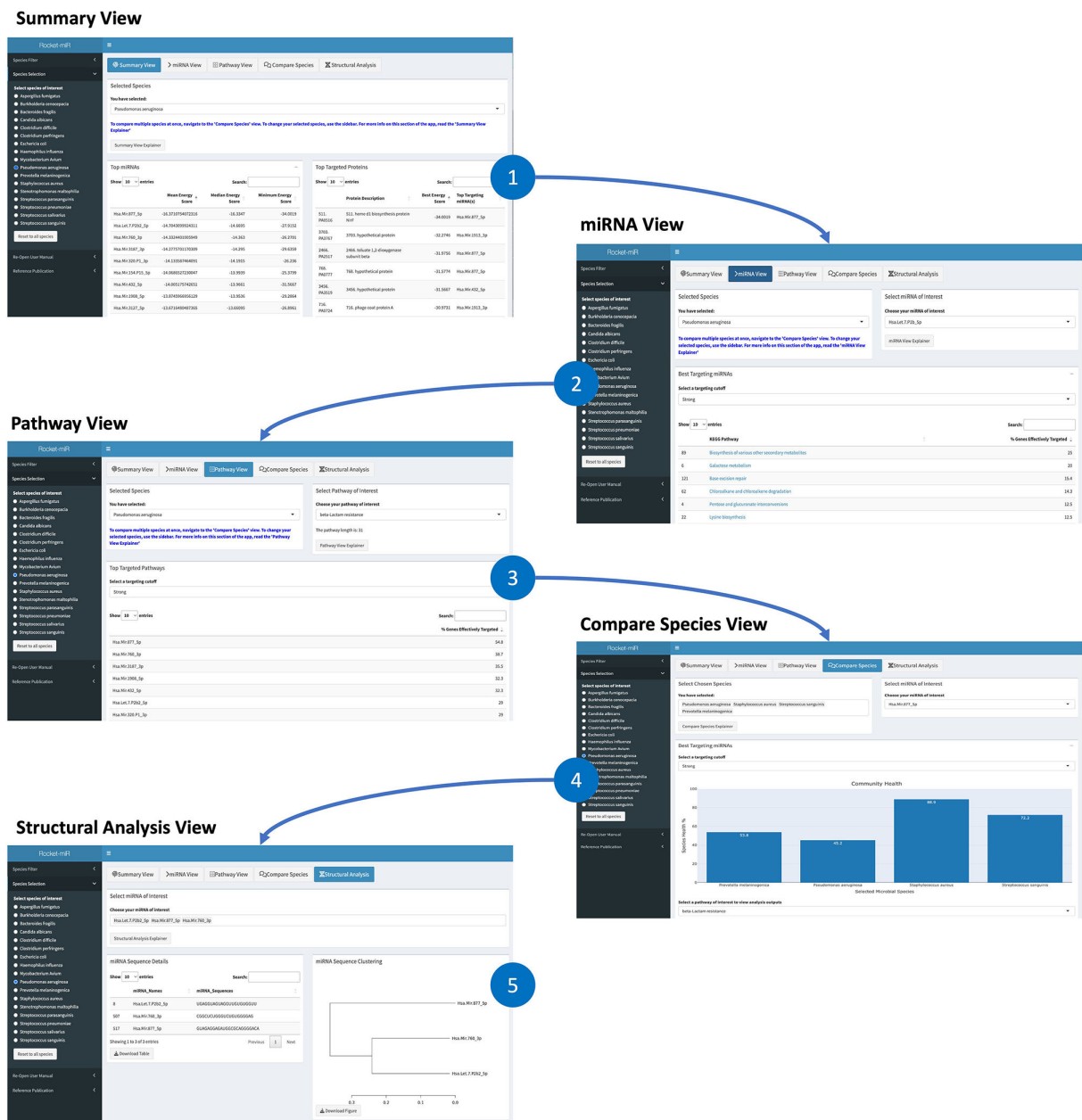

**FIG 1** The Rocket-miR application contains five sections. Enlarged versions of the panels in this figure are available as Fig. S1 to S5. After selecting a pathogen of interest, the user is presented with the Summary View (1), where they can view a summary of the target prediction data for all human miRNAs and all genes of the chosen pathogen. They can then navigate to the other four application views. The miRNA View (2) displays the KEGG pathways best targeted by each individual human miRNA in a chosen pathogen. The Pathway View (3) shows the reverse—the most effective miRNAs for targeting a given pathway. The Compare Species View (4) allows the user to compare the effect of a selected miRNA on different KEGG pathways across species. Finally, the Structural Analysis View (5) allows the user to view and compare the nucleotide structure of all the human miRNAs incorporated in the application.

antimicrobial effects that may target multiple species simultaneously. This section supports the case studies that follow and provides helpful guidance for the use of the Rocket-miR application.

Throughout this and the following sections of the results, readers should keep in mind that the Rocket-miR results are bioinformatic predictions. *In vitro* experiments involving the miRNAs and pathogens included in the application might not yield exactly the results predicted. The miRNA-mRNA target prediction scores in the Rocket-miR application were generated using the target prediction algorithm IntaRNA, which

predicts the strength of interaction based on both the free energy of miRNA-mRNA hybridization and the miRNA secondary structure (58). More negative targeting scores predict stronger miRNA-mRNA interactions. The Rocket-miR application includes three targeting cutoffs (strong: −16.98, medium: −15.35, weak, −13.47), varying from highest to lowest confidence that the miRNA-mRNA interaction will successfully suppress protein expression, based on prior data from our previous let-7b-5p study (38). The IntaRNA algorithm and choice of targeting cutoffs are explained further in Materials and Methods.

A comparative analysis of miRNA target prediction data for all species in the application indicates that some species are likely to be more effectively targeted in general by human miRNAs than others. Approximately 5% of all energy score inter-actions for *A. fumigatus* meet or exceed the medium targeting score cutoff in the application. In contrast, the next best species in terms of overall targeting potential (*P. aeruginosa*, *A. xylosoxidans*, *S. maltophilia, M. abscessus*) have between 2% and 3% of their interactions meeting the medium targeting cutoff. On the lower end, there are three species (*C. perfringens*, *C. difficile*, and *S. aureus*) with less than 1% of interactions meeting the cutoff (Fig. 2).

From Fig. 2, one might conclude that only a small percentage of genes in human pathogens can be effectively targeted by human miRNAs. However, although most human miRNAs are not predicted to target very many human genes, a small number of human miRNAs are predicted to target many genes across multiple human pathogens. This small set of miRNAs therefore has great potential as possible antimicrobial agents. From a single-species perspective, our application identified 92 human miRNAs that are predicted to effectively target 10% or more of all genes in *A. fumigatus* (again at the medium targeting cutoff of −15.35). There are five additional species with 40 or more miRNAs that effectively target 10% or more genes (*M. abscessus*, *P. aeruginosa*, *A. xylosoxidans*, *M. avium*, and *S. maltophilia*) at the same cutoff. All but one species (*S. aureus*) has at least a single human miRNA that targets 10% or more of its genes (Fig. 3A).

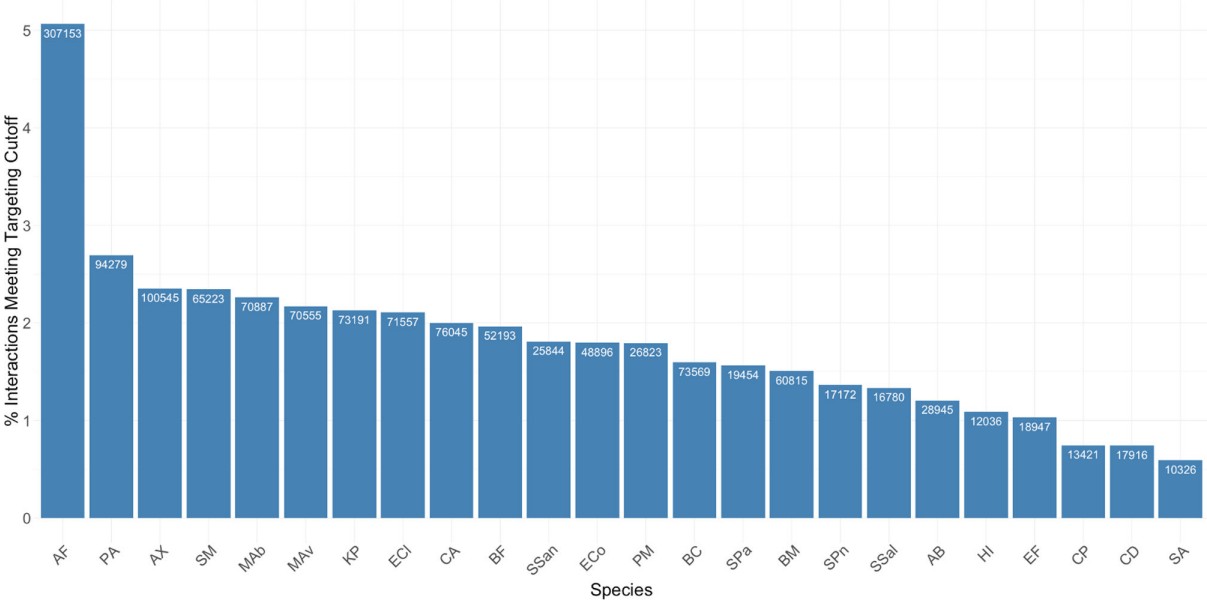

**FIG 2** The overall percentage of miRNA-targeting interactions for each species that meet or exceed the medium targeting cutoff (−15.35) varies between species. Each targeting interaction represents the likelihood of stable interaction between a human miRNA and a particular gene in each human pathogen.. The numbers inside of the bars represent the total number of miRNA-gene interactions that meet the cutoff for each species, whereas the y-axis represents the percentage of interactions meeting the cutoff out of the total number of interactions for each species. Species names are abbreviated as follows: AF, *A. fumigatus*; PA, *P. aeruginosa*; AX, *A. xylosoxidans*; SM, *S. maltophilia*; MAb, *M. abscessus*; MAv, *M. avium*; KP, *K. pneumoniae*; ECl, *E. cloacae*; CA, *C. albicans*; BF, *B. fragilis*; SSan, *S. sanguinis*; ECo, *E. coli*; PM, *P. melaninogenica*; BC, *B. cenocepacia*; SPa, *S. parasanguinis*; BM, *B. multivorans*; SPn, *S. pneumoniae*; SSal, *S. salivarius*; AB, *A. baumannii*; HI, *H. influenza*; EF, *E. faecium*; CP, *C. perfringens*; CD, *C. difficile*; SA, *S. aureus*.

Building on this single-species analysis, we considered whether some human miRNAs are predicted to target multiple pathogenic species simultaneously. Figure 3B identifies the miRNAs most likely to be effective in multiple pathogens. These broadly effective miRNAs include let-7b-5p, miR-877–5p, miR-3127–5p, miR-3138–3p, miR-320-P1-3p, and miR-320-P3-3p. Each of these six human miRNAs are predicted to target 10% or more of genes in at least 20 of the 24 pathogens included in the Rocket-miR application (at the medium cutoff). Table 2 further demonstrates, for each of the six broad-spectrum miRNAs, what proportion of all genes are targeted in each of the application's 24 different species.

Drawing on the combined information from Fig. 2 and 3 ; Table 2, a researcher might make an educated decision about whether targeting a given species with human

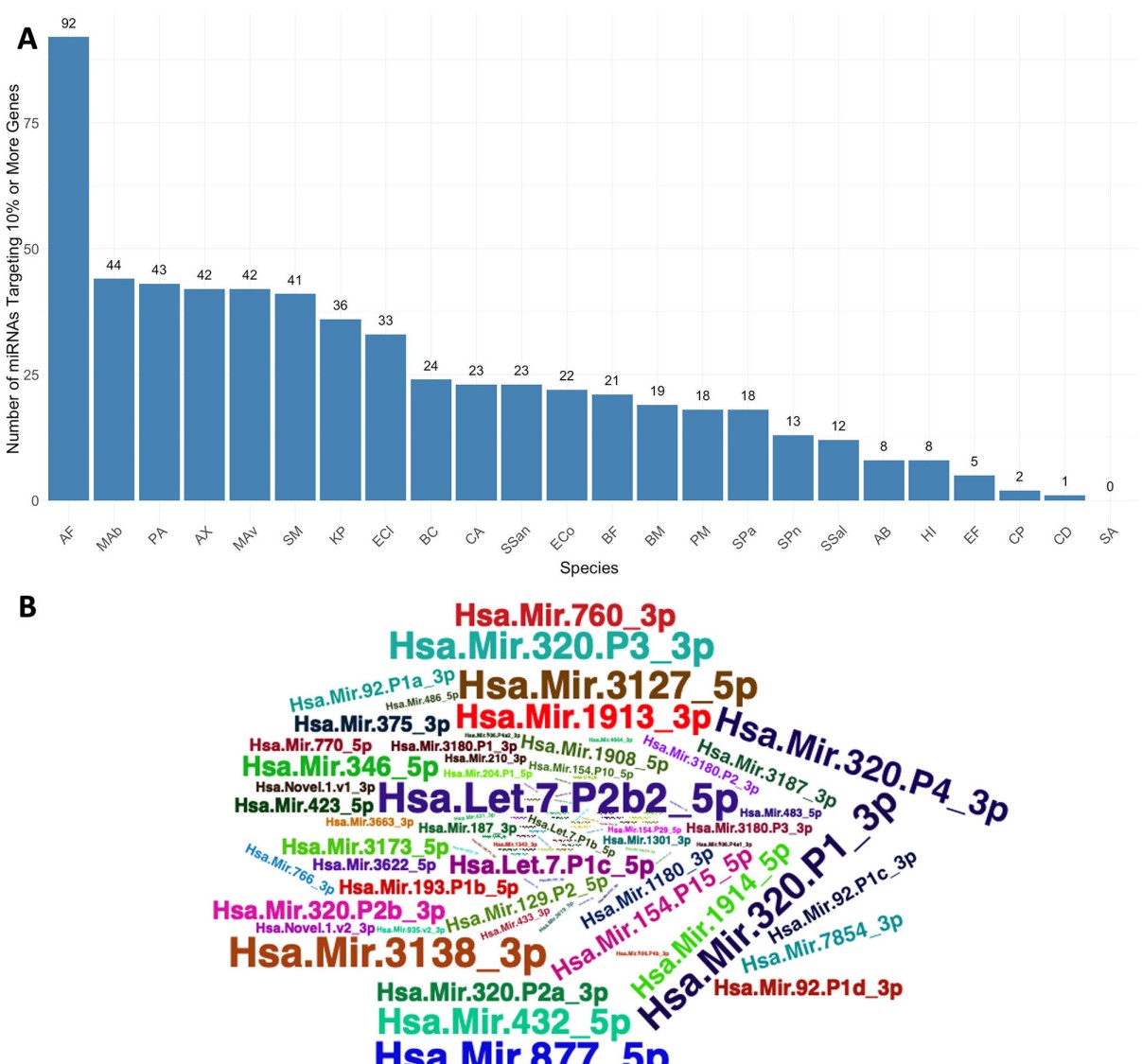

**FIG 3** Identification of broadly effective miRNAs. (A) The number of human miRNAs that target 10% or more genes at or above the medium targeting cutoff (−15.35) varies between species. The height of the bars (and the numbers above the bars) represent the number of miRNAs that meet this criterion for each species. The cutoff of 10% was chosen as a simple measure to predict overall effectiveness of human miRNAs in different species. The Rocket-miR application can be used to predict how effectively individual miRNAs might target specific genes and pathways in each species. (B) Some miRNAs target 10% or more genes at or above the medium targeting cutoff (−15.35) in many different species. The word cloud provides a visual indication of which miRNAs have the best broad-spectrum effects—with miRNAs of a larger text size targeting 10% or more genes at the medium cutoff in a greater number of species. Species abbreviations in the upper panel are the same as Fig. 2.

**TABLE 2** Six miRNAs that target 10% or more genes at the medium targeting cutoff in a combined 24 different CF pathogens[a]

| Species | let-7b-5p | 877-5p | 3127-5p | 3138-3p | 320-P1-3p | 320-P3-3p |
|---|---|---|---|---|---|---|
| *Achromobacter xylosoxidans* | 0.31 | 0.51 | 0.26 | 0.16 | 0.23 | 0.19 |
| *Acinetobacter baumannii* | 0.19 | 0.24 | 0.12 | 0.14 | 0.11 | 0.14 |
| *Aspergillus fumigatus* | 0.53 | 0.78 | 0.45 | 0.56 | 0.51 | 0.52 |
| *Bacteroides fragilis* | 0.21 | 0.33 | 0.14 | 0.20 | 0.10 | 0.14 |
| *Burkholderia cenocepacia* | 0.26 | 0.30 | 0.12 | 0.11 | 0.14 | 0.10 |
| *Burkholderia multivorans* | 0.22 | 0.30 | 0.12 | 0.11 | 0.12 | 0.09 |
| *Candida albicans* | 0.35 | 0.32 | 0.20 | 0.20 | 0.11 | 0.17 |
| *Clostridium difficile* | 0.06 | 0.04 | 0.02 | 0.03 | 0.01 | 0.03 |
| *Clostridium perfringens* | 0.05 | 0.04 | 0.02 | 0.03 | 0.01 | 0.02 |
| *Enterobacter cloacae* | 0.24 | 0.45 | 0.15 | 0.20 | 0.19 | 0.18 |
| *Enterococcus faecium* | 0.10 | 0.17 | 0.06 | 0.11 | 0.05 | 0.08 |
| *Escherichia coli* | 0.21 | 0.40 | 0.13 | 0.17 | 0.15 | 0.16 |
| *Haemophilus influenza* | 0.15 | 0.21 | 0.12 | 0.14 | 0.10 | 0.12 |
| *Klebsiella pneumoniae* | 0.24 | 0.49 | 0.14 | 0.20 | 0.22 | 0.22 |
| *Mycobacterium abscessus* | 0.25 | 0.39 | 0.17 | 0.16 | 0.20 | 0.16 |
| *Mycobacterium avium* | 0.29 | 0.35 | 0.17 | 0.13 | 0.21 | 0.18 |
| *Prevotella melaninogenica* | 0.25 | 0.36 | 0.13 | 0.25 | 0.13 | 0.18 |
| *Pseudomonas aeruginosa* | 0.41 | 0.61 | 0.27 | 0.27 | 0.35 | 0.29 |
| *Staphylococcus aureus* | 0.09 | 0.06 | 0.04 | 0.04 | 0.02 | 0.03 |
| *Stenotrophomonas maltophilia* | 0.32 | 0.44 | 0.21 | 0.15 | 0.23 | 0.19 |
| *Streptococcus parasanguinis* | 0.19 | 0.34 | 0.16 | 0.19 | 0.18 | 0.21 |
| *Streptococcus pneumoniae* | 0.18 | 0.30 | 0.13 | 0.17 | 0.15 | 0.19 |
| *Streptococcus salivarius* | 0.19 | 0.30 | 0.13 | 0.18 | 0.15 | 0.18 |
| *Streptococcus sanguinis* | 0.19 | 0.37 | 0.15 | 0.20 | 0.18 | 0.21 |

[a]The exact percentage of genes targeted in each pathogen is listed for each of the five miRNAs.

miRNAs is an endeavor worthy of laboratory resources, and if so, what miRNAs would be most promising to consider as antimicrobial candidates. Though predicted targeting of a large portion of genes in the genome is not a guarantee that a miRNA will effectively interfere with the growth and/or virulence capabilities of a given pathogen, it does increase the likelihood that the miRNA will have a potent effect.

## Experimental validation of Rocket-miR predictions

To validate Rocket-miR predictions with *in vitro* experiments, the effect of let-7b-5p expression on the planktonic growth and biofilm formation of *P. aeruginosa* was performed using biofilm inoculator (peg lid) plates to measure colony forming units (CFUs; as described elsewhere) (59). An expression vector containing the let-7b-5p sequence was electroporated into *P. aeruginosa* to induce let-7b-5p expression, as described in Materials and Methods. As shown in Fig. 4A, let-7b-5p expression reduced planktonic growth of *P. aeruginosa* in the presence and absence of aztreonam (1/2 the minimum inhibitory concentration), an antibiotic commonly used to treat people with CF, compared to empty vector control with and without aztreonam (60). Figure 4B demonstrates that let-7b-5p inhibited biofilm formation of *P. aeruginosa* in the presence of aztreonam compared to empty vector control plus aztreonam (Fig. 4B).

These new results are in line with previous experiments. In a prior study, we had demonstrated, using a crystal violet assay, that addition of aztreonam (1/2 the minimum inhibitory concentration) and expression of let-7b-5p in *P. aeruginosa* had an additive effect to reduce biofilm formation (38, 61). The additive effect of let-7b-5p and aztreonam is predicted based on the observation that let-7b-5p inhibits numerous genes in the biofilm formation pathways and inhibits aztreonam efflux pumps in *P. aeruginosa* (38).

These experimental results confirm the Rocket-miR prediction that let-7b-5p effectively targets *P. aeruginosa*, with significant impacts on growth (planktonic and

biofilm) and antibiotic sensitivity. They also demonstrate a simple approach for assessing the effects of human miRNAs on microbial cells that can be leveraged in future studies with different miRNAs, microorganisms, and antibiotics. To provide additional inspiration for future studies, the following three case studies explore how the Rocket-miR application can be used to predict the effects of human miRNAs on the level of individual genes and biological pathways in selected bacteria. The case studies cover several clinically relevant human pathogens.

### Case study #1: identifying antimicrobial miRNAs to target β-lactam resistance in *Pseudomonas aeruginosa*

#### Sharanya Sarkar, Ph.D. candidate, Stanton Lab, Geisel School of Medicine

Although the CF lung is characterized by polymicrobial infections involving a wide variety of species such as *Staphylococcus aureus*, *Haemophilus influenzae*, and *Burkholderia cenocepacia*, 50% of adult pwCF are colonized by *Pseudomonas aeruginosa* (*P.a.*) (62, 63). Prolonged colonization by *P.a.* is linked with worse lung function and enhanced mortality rates in pwCF (64). In the chronic state, *P.a.* escapes immune clearance and resists conventional antibiotics via mechanisms such as the production of β-lactamases to degrade antibiotics and efflux pumps to export antibiotics from *Pseudomonas* cells, reduced expression of porin channels to prevent antibiotic intake, and adopting a biofilm

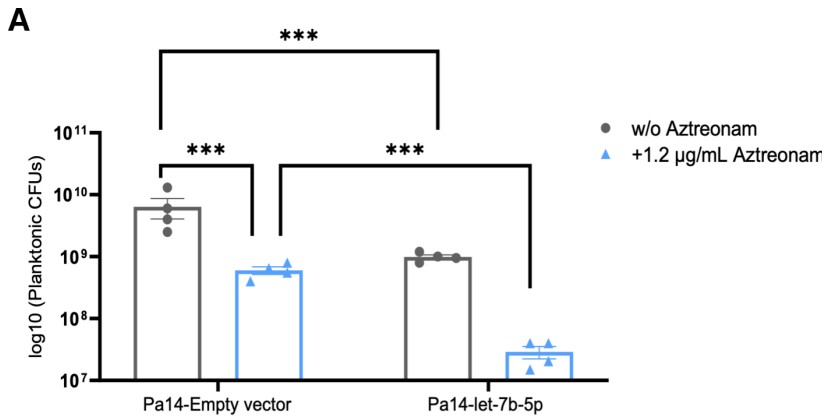

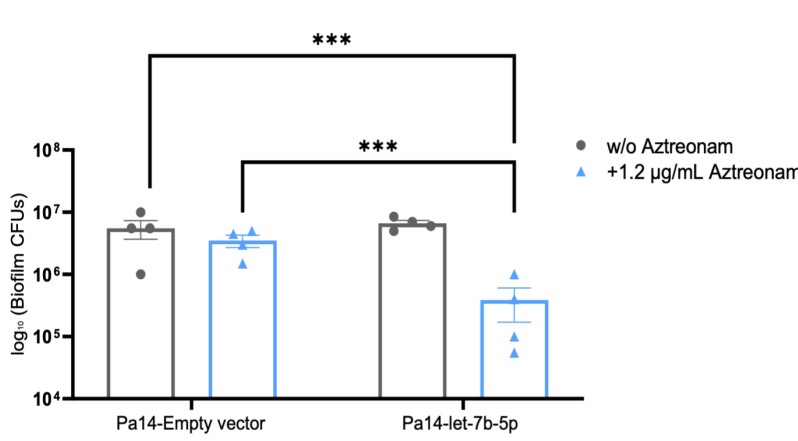

**FIG 4** Let-7b-5p reduced planktonic growth and biofilm formation of *P. aeruginosa* as predicted by Rocket-miR. (A) *P. aeruginosa* transfected with let-7b-5p exhibited less planktonic growth compared to *P. aeruginosa* transfected with the empty plasmid in the presence and absence of aztreonam. (B) Let-7b-5p also significantly reduced biofilm formation by *P. aeruginosa* in the presence of aztreonam compared to empty vector in the presence of aztreonam. ***$P < 0.0001$. Experiments were repeated four times.

community structure that is less susceptible to antibiotic treatment (65). Thus, new treatments are required to reduce the *P.a.* burden in the CF lung by targeting pathways such as antibiotic resistance and biofilm formation. Accordingly, the Stanton lab is developing miRNA-based antimicrobial therapies to eliminate *P.a.* and other pathogens in the CF lung.

Leveraging the analysis capabilities of this application, we were particularly interested in identifying novel therapeutic miRNAs that could target antibiotic resistance and biofilm formation pathways in *P.a.* After selecting *Pseudomonas aeruginosa* from the "Species Selection" tab in the sidebar, the "Summary View" arranged the human miRNAs in the order of how effectively they targeted genes in *P.a.* We found that miR-877–5p had the best median energy score—indicating that it was predicted to effectively target more genes in *P. aeruginosa* than any other human miRNA. From the "miRNA View" section of the application, we downloaded an excel file with the complete list of *P.a.* KEGG pathways and the corresponding percentage of genes targeted by miR-877-5p. Since we were most interested in virulence pathways such as biofilm formation and antibiotic resistance, we examined pathways associated with these terms and found that 46.0% of genes in the biofilm formation pathway and 54.8% of genes in the β-lactam resistance pathway were predicted to be effectively targeted by miR-877-5p. These values were obtained by selecting a "Strong" targeting cutoff, thereby giving us the maximum confidence in the likelihood of protein targeting by this miRNA. To understand exactly what genes in the two pathways were predicted to be targeted by miR-877-5p, we clicked on the hyperlinked pathway terms in the miRNA View. These hyperlinks opened two new windows on the browser depicting the targeted genes on a KEGG pathway diagram (Fig. 5A and B). To further establish the targeting power of miRNA-877-5p vs other human miRNAs, we used the "Pathway View" section of the application and selected biofilm formation and β-lactam resistance as our chosen pathways. The Pathway View indicated that miRNA-877-5p was indeed the human miRNA predicted to target these two pathways most effectively, with the second most effective miRNA targeting as much as 15% fewer genes in both cases.

Since much of our prior work in the lab has focused on let-7b-5p, we wanted to run an analysis to compare the predicted effects of miR-877-5p and let-7b-5p and consider miR-877-5p as a new candidate for future therapeutic developments. We performed the same analysis as outlined above in this case study for let-7b-5p and summarized our findings (Table 3). While let-7b-5p had a weaker performance as compared to miR-877-5p on most of the key parameters, the two miRNAs targeted complementary genes in some sub-pathways such as the chemosensory system, *Pseudomonas* quinolone signal(PQS) system, and β-lactamase classes. For instance, while let-7b-5p showed predicted targeting for the class C β-lactamase, miR-877-5p showed predicted targeting for the class D β-lactamase.

This case study led to the development of the hypothesis that human microRNA 877-5p can be used as a potent antimicrobial candidate, specifically against the notorious CF pathogen *Pseudomonas aeruginosa*. This hypothesis can be easily translated to wet bench experiments, which in turn, will play an important role in understanding host-pathogen interactions in the context of eukaryotic miRNA regulation. For instance, naturally occurring EVs or synthetic polymeric nanoparticles, loaded with miR-877-5p, could be administered in CF mice models to test their efficacy in abating *P.a.* infection and inflammation. There is prior precedent for the study of miR-877-5p as a therapeutic agent. For example, miR-877-5p has been shown by several research groups to inhibit hepatocellular, prostate, lung, and gastric cancers (66–69). Furthermore, miR-877-5p has also been demonstrated to alleviate acute respiratory distress syndrome both *in vitro* and *in vivo* (70). Therefore, utilizing this miRNA in the context of CF pathology would be a logical step. To achieve even better results, one might combine let-7b-5p with miR-877-5p in EVs or nanoparticles to achieve complementary gene targeting in multiple pathways. One might also co-load vesicles with a β-lactam antibiotic and miRNA-877-5p to improve the efficacy of the antibiotic while the β-lactam resistance

**TABLE 3** Comparative analyses of key parameters between miRNAs let-7b-5p and miR-877-5p

| Key parameters | microRNA | |
|---|---|---|
| | let-7b-5p | miR-877-5p |
| Median interaction energy score for *P.a.* | 14.67 | 16.33 |
| % Genes targeted in biofilm formation pathway | 31.0% | 46.0% |
| % Genes targeted in β-lactam resistance pathway | 29.0% | 54.8% |
| Proteins targeted in biofilm formation pathway | PilJ, ChPa., CyaB, FleQ, PqsE, PA1611, PA1976, GacS, RetS, HSI-I, TpbB, RoeA, MucR, BifA, FleQ, Psl, FunX, FunW | PilI, ChPa, PilG, CyaB, CpdA, FleQ, ExsA, PqsA, MvtR, RhlR, RhlC, LecB, LasI, SagS, PA1976, HsbR, HsbA, LadS, RetS, HSI-I, WspB/D, Wsp-F, TpbB, RoeA, MucR, BifA, FleQ, Psl, PelB, PelD, PelE, PelF, PelG, FunX, FunW |
| Proteins targeted in β-lactam resistance pathway | OPaD, RND, Class C β-lactamase, AmpG, FtsI, NalC, AmpR, OprD, MexB/AcrB | OPaD, OMP, RND, MFP, Class D β-lactamase, AmpG, ParS, ParR, OprD, FtsI, NalC, MexZ, MexX, MexY, MexR, NalD, MexB/AcrB |

pathway is rendered ineffective by the miRNA. The total analysis in this application took around 30 min, and outstanding features such as typing in direct terms in the search boxes (rather than scrolling through a long list) made the process even faster. Overall, Rocket-miR was a fast and user-friendly experience, and helped us to develop a testable hypothesis with results that could advance miRNA-based interventions as a therapeutic strategy for CF.

## Case study #2: targeting a polymicrobial community of CF pathogens

*Sharanya Sarkar, Ph.D. candidate, Stanton Lab, Geisel School of Medicine*

Building on the first case study, we wanted to explore the potential use of miR-877-5p to target four key CF pathogens simultaneously, namely, *Prevotella melaninogenica*, *Pseudomonas aeruginosa*, *Staphylococcus aureus*, and *Streptococcus sanguinis* (Fig. 6). These four pathogens constitute a familiar CF "pulmotype" based on factors from prior research such as prevalence/abundance, co-occurrence, physical/spatial proximity, metabolic cross-feeding, and capacity to dictate health outcomes in pwCF (71, 72). *P.a.*

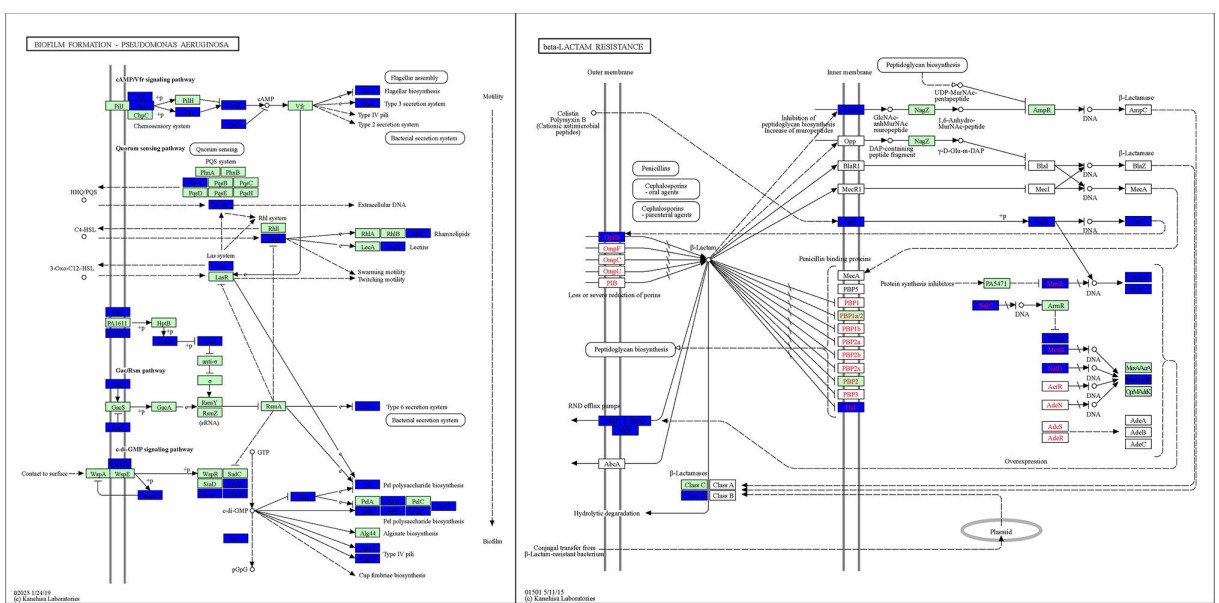

**FIG 5** Key genes targeted by human 877-5p in (i) the biofilm formation pathway and (ii) the β-lactam resistance pathway in *Pseudomonas aeruginosa* are highlighted in blue. Highlighted genes meet the application's strong targeting cutoff, with a predicted energy score less than or equal to −16.98 (a more negative energy score being indicative of stronger predicted targeting). Green genes are present in the *P. aeruginosa* genome but not strongly targeted. The rationale for the values of the targeting cutoffs is described further in Materials and Methods.

has been shown to significantly influence the inflammatory responses, independent of *S. aureus* in chronic infection models, and also double the risk of death in adult pwCF (73). *S. aureus*, on the contrary, is usually more prevalent in younger, healthier CF patients (74). *Prevotella* is a common anaerobe found in the CF lung, and a study has reported that anaerobes like *Prevotella* detected in the sputum during periods of clinical pulmonary exacerbation were linked with lessened inflammation and better lung function, with respect to sputum samples containing *P.a.* (75). Therefore, reducing *P.a.* abundance relative to *S. aureus* and *P. melaninogenica* may improve clinical outcomes.

Expanding on these results, it may be possible to target the β-lactam resistance pathway in this community of species more effectively by leveraging multiple human miRNAs simultaneously. Figure 7 demonstrates the combined effect of the five broad-spectrum miRNAs identified in Table 2 against the β-lactam resistance pathway for the four-species community as an example of how miRNAs may be combined. More genes on the β-lactam resistance pathway are effectively targeted when the six miRNAs are used together than when miR-877-5p is used alone—with one-third of the *S. aureus* genes on the β-lactam resistance pathway and more than 50% of the *S. sanguinis* genes on the pathway predicted to be targeted (compared to ~10% and ~25% for miR-877-5p alone). Targeting of *P. melaninogenica* and *P. aeruginosa* is also predicted to improve, with approximately 75% and 80% of genes targeted, respectively (compared to ~45% and ~55%, respectively).

## Case study #3: human miRNAs target the *Bacteroides fragilis* type VI secretion system (T6SS)

### *Casey Latario, Ph.D. candidate, Ross Lab, Geisel School of Medicine*

In this final case study, we sought to gain insight into the potential for host-microbe interactions mediated by miRNA targeting of a bacterium inhabiting a different site in the body, the colonic symbiont *Bacteroides fragilis*. *B. fragilis* is present exclusively in mammals and is both prevalent and relatively abundant in the healthy human gastrointestinal microbiome (76, 77). *B. fragilis* is generally considered to be a beneficial organism, though enterotoxigenic strains can be associated with inflammatory intestinal disease and colonic tumorigenesis (78). In pwCF, *Bacteroides* species including *B. fragilis* are diminished in abundance compared to healthy controls and there is great interest

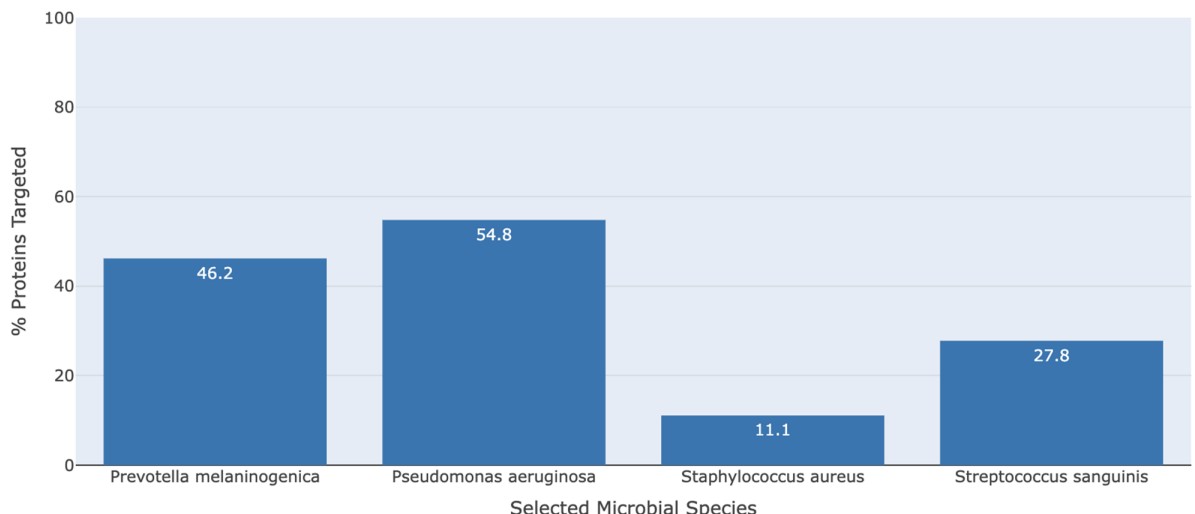

**FIG 6** Community health bar graph demonstrating the β-lactam resistance pathway targeting of miRNA 877-5p in four key CF species—*Prevotella melaninogenica, Pseudomonas aeruginosa, Staphylococcus aureus,* and *Streptococcus sanguinis.* In this figure, downloaded directly from the Rocket-miR application, 100% protein targeting would indicate that all genes on the β-lactam resistance pathway are effectively targeted by a given miRNA. β-Lactam resistance is targeted most effectively by miRNA 877-5p in *P.a.* (54.8% of genes targeted) and least in *S. aureus* (11.1% of genes targeted).

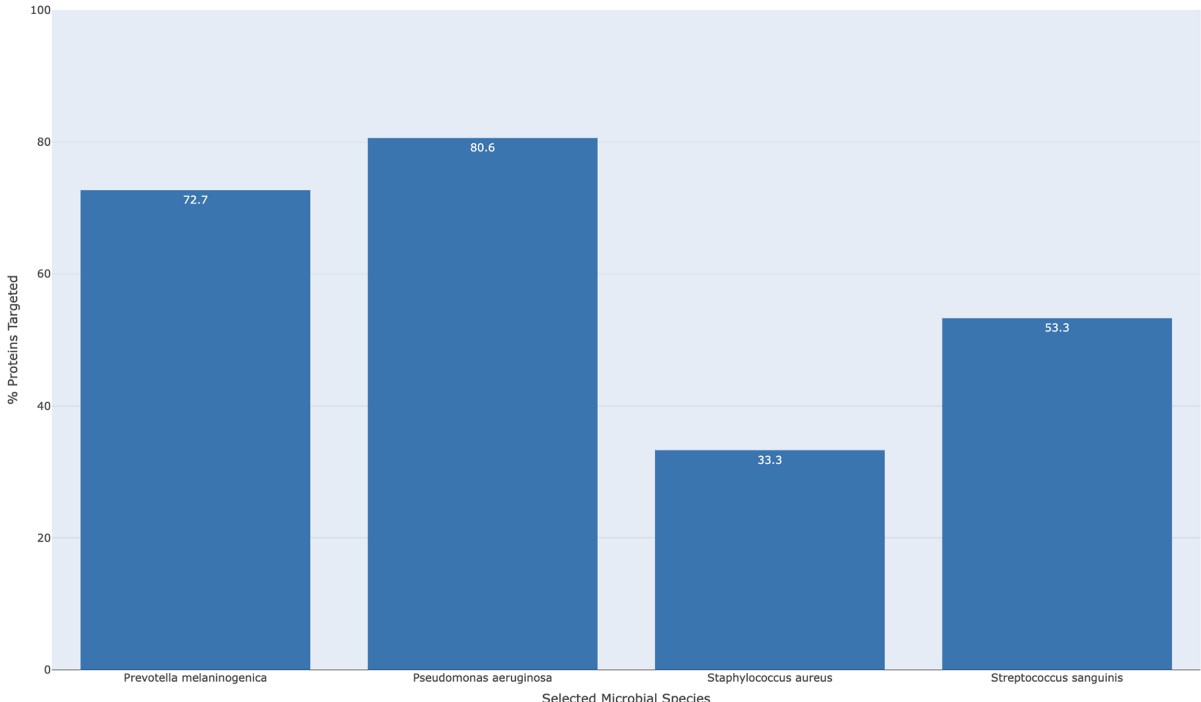

**FIG 7** The combined predicted effect of the six broad-spectrum human miRNAs shown in Table 3 (let-7b-5p, 3127-5p, 3138-3p, 320-P1-3p, 320-P3-3p, 877-5p) on β-lactam resistance in this CF-relevant four-species microbial community is greater than the effect of the individual miRNAs 877-5p or let-7b-5p administered alone (Fig. 3; Table 2). As in Fig. 3, the most stringent cutoff in the application (−16.98) was used to determine which genes on the β-lactam resistance KEGG pathway were targeted effectively in each species. The height of the bars (and values inside the bars) represents the proportion of total genes in the β-lactam resistance KEGG pathway where at least one of the six miRNAs meet or exceed the targeting cutoff.

in identifying avenues for the restoration of these populations back to healthy levels to alleviate gastrointestinal symptoms in CF (79, 80).

We used Rocket-miR to identify miRNA targets in the genome of the reference strain of *B. fragilis* NCTC 9343. The top hit predicted by Rocket-miR was 1913_3p, which targets *tssG* mRNA with an energy score of −30.97 (Table 4). TssG is an integral part of the T6SS, a nanomachine first discovered in *Vibrio cholerae* and *Pseudomonas aeruginosa,* but later found to be broadly distributed throughout Gram-negative bacteria (81–83). The T6SS functions to translocate toxic effector proteins in a contact-dependent manner to a recipient target cell (84). Since translocated effector proteins often possess endonuclease, protease, phospholipase, or other activities, their concerted action typically results in bacteriostasis or bacteriolysis of targeted cells, thereby providing a fitness advantage to the producing bacterium (85). The T6SS is composed of three subcomplexes: the membrane complex (MC), baseplate (BP), and tail-tube complex (TTC) (86). As yet only functionally characterized in Proteobacteria, TssG is an essential conserved component of the baseplate subcomplex, which is composed of six heterotetrameric "wedges" (87). TssG forms the core of each wedge (Fig. 8). Deletion of *tssG* inhibits assembly of the baseplate and thus ablates T6SS activity. We predict that the outcome of miR-1913_3p inhibition of *tssG* would be the reduction of *B. fragilis* T6SS activity in the gut.

Notably, in addition to *tssG*, we found that many other *B. fragilis* T6SS genes were predicted by Rocket-miR to be the targets of human-encoded miRNAs with strong predicted energy scores. These genes included two other BP genes in addition to TssG, three MC genes, two TTC genes, the T6SS-associated ClpV ATPase, and two effector immunity gene pairs (Table 4). Inactivation of any of these genes could be inferred to impair or inactivate the T6SS. In sum, 16/26 (61.5%) *B. fragilis* T6SS genes are predicted to be effective targets for human miRNAs.

**TABLE 4** Sixteen *B. fragilis* T6SS genes and their predicted targeting by human miRNAs[a]

| Locus tag | T6SS nomenclature | Subcomplex | miRNA | Energy score |
|---|---|---|---|---|
| BF9343_RS09380/BF9343_1923 | TssG | BP | Hsa.Mir.1913_3p | −30.9755 |
| BF9343_RS09390/BF9343_1925 | TssN | MC | Hsa.Mir.3138_3p | −23.7509 |
| BF9343_RS09415/BF9343_1930 | VgrG | TTC | Hsa.Mir.504.v2_5p | −22.4218 |
| BF9343_RS09360/BF9343_1919 | TssR | MC | Hsa.Mir.877_5p | −21.4969 |
| BF9343_RS09410/BF9343_1929 | PAAR | TTC | Hsa.Mir.3944_3p | −21.2825 |
| BF9343_RS09465/BF9343_1940 | ClpV | ATPase | Hsa.Mir.185_5p | −21.2479 |
| BF9343_RS09385/BF9343_1924 | TssK | BP | Hsa.Mir.3138_3p | −20.2922 |
| BF9343_RS09420/BF9343_1931 | TssF | BP | Hsa.Mir.877_5p | −20.0846 |
| BF9343_RS09430/BF9343_1933 | TagC | Unknown | Hsa.Mir.4664_3p | −19.5252 |
| BF9343_RS09405/BF9343_1928 | bte2 | Effector | Hsa.Mir.877_5p | −19.2061 |
| BF9343_RS09400/BF9343_1927 | bti2a | Immunity | Hsa.Let.7.P1b_5p | −18.5971 |
| BF9343_RS09485/BF9343_1944 | TetR | Unknown | Has.Mir.130.P4a_5p | −18.2097 |
| BF9343_RS09365/BF9343_1920 | TssP | MC | Hsa.Mir.204.P1_5p | −18.1975 |
| BF9343_RS09450/BF9343_1937 | bte1 | Effector | Hsa.Mir.139_5p | −18.1686 |
| BF9343_RS09460/BF9343_1939 | Hcp | TTC | Hsa.Mir.432_5p | −17.4228 |
| BF9343_RS09395/BF9343_1926 | bti2b | Immunity | Hsa.Mir.430.P24_3p | −16.8882 |

[a]The *B. fragilis* NCTC 9343 strain locus tags are indicated for each gene, alongside the T6SS gene designation, the subcomplex to which each corresponds, and the Rocket-miR energy score for the human miRNA predicted to target the gene most effectively. A more negative energy score corresponds to a stronger interaction.

Considering these data, we predict that exposure of *B. fragilis* cells to EVs released by colonic cells containing miR-1913_3p or other miRNAs listed in Table 4 would result in reduced T6SS expression and subsequent diminishment of interbacterial antagonism. In addition, engineered extracellular vesicles or nanoparticles containing these miRNAs could be developed to reduce T6SS and therefore reduce the pathogenicity of *B. fragilis*. Since EVs secreted by colon cells are likely to exist in a gradient extending luminally from the epithelial layer, one potential function of this mechanism could be the

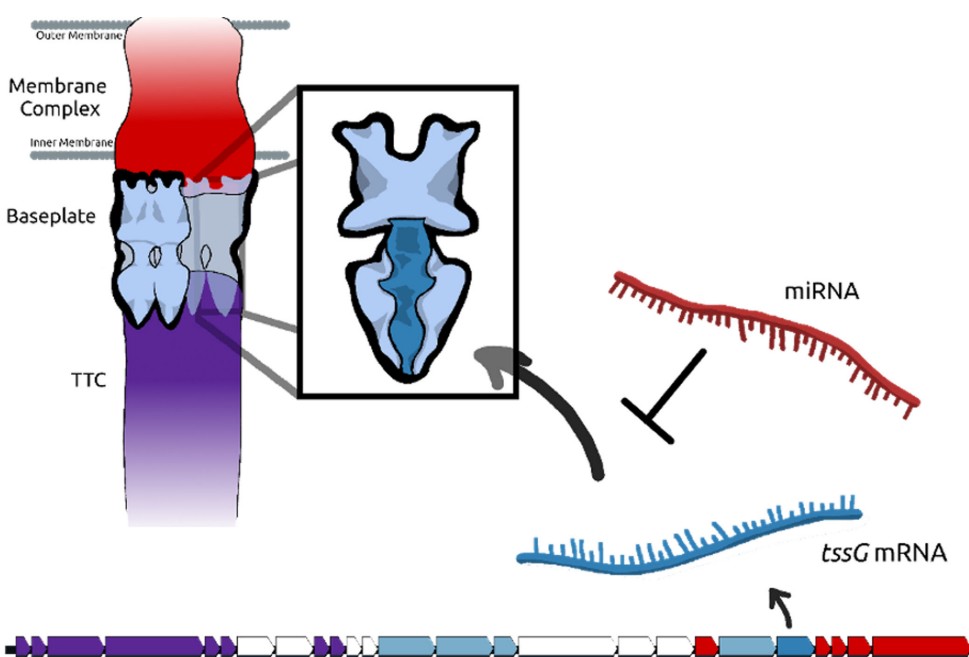

**FIG 8** Rocket-miR identified the type VI secretion system gene *tssG* harbored by *Bacteroides fragilis* as the top predicted gene targeted by human miRNA. The schematic depicts a T6SS, composed of three distinct subcomplexes: the MC, BP, and TTC. The T6SS BP is formed by units of TssEFGK wedges (inset), with TssG as the central protein (indicated in dark blue). Below the T6SS is a *B. fragilis* T6SS locus, with the *tssG* gene shown in dark blue. The transcribed *tssG* (blue mRNA) is shown interacting with a potential inhibitory miRNA, 1913_3p (red miRNA). MC proteins and genes are indicated in red, TTC in purple.

dampening of T6SS-dependent bacterial lysis near the epithelium to prevent unwanted inflammatory responses to bacterial-derived cellular products. This dampening would also be predicted to lower *B. fragilis* fitness during interactions with other Gram-negative bacterial competitors, but may promote co-existence especially with other *Bacteroides* species (88, 89). While the mucosal layer acts to buffer the epithelium from the bulk of the gut microbiota, *B. fragilis* can colonize colonic crypts and even form polymicrobial biofilms at sites of colonic polyp formation (90, 91) . How or if *B. fragilis* biogeography and interbacterial interactions are impacted in CF is currently unknown. Further mechanistic investigation of the impact of miRNAs on *B. fragilis* T6SS in both healthy individuals and in the context of CF will be important to lend experimental credence to our observations and illuminate potential therapeutic avenues.

## DISCUSSION

By designing Rocket-miR as an accessible, point-and-click web application, we have made the results of miRNA target prediction accessible to a broad audience of researchers. Researchers at any level of computational experience can access target prediction data in Rocket-miR within a few minutes. Without such an application, it would take weeks for an experienced bioinformatician with access to high performance computing resources to gather and analyze the same target prediction data for just a single pathogen. While miRNA target prediction databases do exist (92, 93), they are limited to model organisms and other commonly studied species—not most microbial pathogens such as those that infect people with CF and other diseases. To our knowledge, Rocket-miR is the only data mining tool of its kind that exists for the microbiology community. The Rocket-miR application provides this research community a starting point to develop a new class of miRNA-based antimicrobial therapeutics. Drawing on public genomic sequencing data in bacteria, it aims to encourage new wet bench experiments that will expand the range of treatment options for drug-resistant pathogens. These experiments will also enhance biological understanding of host pathogen miRNA-mRNA interactions.

As our case studies demonstrate, scientists may use human miRNAs to target individual genes and biological pathways composed of multiple functionally related genes. Targeting KEGG pathways related to antibiotic resistance, biofilm formation, or other virulence traits could help treat drug-resistant infections. Furthermore, while small molecule drugs are traditionally designed to target individual proteins, miRNA-based therapeutics would likely have a much broader effect, disrupting the function of many genes simultaneously. While a bacterium may evolve resistance to a small molecule that targets an individual protein, it is much less likely that this bacterium could evolve to resist an miRNA that is targeting many genes at once (55–57). An effective therapeutic regimen could even combine miRNA-based drugs with more precise small molecule drugs or traditional antibiotics. As a similar example, engineered vesicles or nanoparticles have been designed for cancer treatment that include both small RNAs or small RNA antagonists and chemotherapeutic agents. In addition to the synergistic effects they enable, these delivery vehicles also make the small RNAs more effective by protecting them from ribonucleases (94–96).

The Rocket-miR application encourages researchers to consider not only the possibility of targeting individual pathogens, but also targeting multiple pathogens simultaneously. We have identified a set of miRNAs with predicted broad-spectrum effects across multiple CF pathogens. This includes let-7b-5p, which we have found in previous research to effectively target virulence pathways in *Pseudomonas aeruginosa*. We have further predicted that let-7b-5p will be effective against a community of species including *P. aeruginosa*, *P. melaninogenica*, *S. aureus*, and *S. sanguinis*.

This analysis extends our understanding of human miRNA-pathogen mRNA interactions. Of the 24 CF pathogens included in the Rocket-miR application, we predict that the fungal species *A. fumigatus* would be most effectively targeted by human miRNAs—with approximately twice as many targeting interactions meeting the application's medium targeting cutoff than the next best targeted species. An interesting

topic of further study would be to explore the underlying reasons for the differences in predicted targeting across species shown in Fig. 2 and 3. Making and testing target predictions for additional species and/or strains of a bacterium may uncover the factors that underlie a pathogen's susceptibility to targeting by human miRNAs. For example, it may be that the relative evolutionary closeness of *Aspergillus* and other fungal species to humans (i.e., both are eukaryotes) renders them more susceptible to targeting by human miRNAs in general than bacterial species. It is also possible that genomic features like GC content make a given pathogen more or less susceptible to targeting by human miRNAs.

The pathogen that is predicted to be least effectively targeted by human miRNAs is the common Gram-positive bacterium *S. aureus*, with no human miRNAs predicted to target more than 10% of its genes (Fig. 3A). However, it may be possible to treat less susceptible pathogens like *S. aureus* with a combination of human miRNAs to increase the total number of genes (and biological pathways) that can be effectively targeted (Fig. 7). Investigations along these lines would be worthwhile, as antibiotic-resistant *S. aureus* infections are common in people with CF as well as in hospital-acquired infections (97, 98). Alternatively, as another way to target *S. aureus* and other CF pathogens as well, it may be possible to design miRNA mimics that are based on human miRNAs but altered in sequence to improve targeting success (99, 100). Researchers could combine IntaRNA target predictions with Rocket-miR's visualization to compare the predicted effects of their own designed miRNA mimics on *S. aureus* or other human pathogens.

Though a demonstrably useful tool, the Rocket-miR application does have limitations. One drawback is that the application includes data on only one strain of each human pathogen, as we chose to prioritize including a diverse range of different pathogens rather than multiple strains of a smaller set of pathogens. Two strains of the same species are likely to have a large degree of genetic homology and consequently similar miRNA-targeting prediction data. In support for this assumption, several pairs of species from the same genus (e.g., *B. cenocepacia* and *B. multivorans; C. difficile* and *C. perfringens*) are included in the application and have very similar miRNA-targeting profiles. However, when comparing two strains, one may possess genes that the other lacks, and the sequence homology for a given gene will not be perfect, causing target predictions to differ. Future investigations of strain-specific miRNA targeting would be worthwhile.

Another concern is challenges related to miRNA-based drug development, including side effects and delivery. As noted in the Introduction, a range of "miRNA mimics" (synthetic compounds that mimic the structure of miRNAs with known therapeutic properties) have been designed and tested in clinical trials for various human cancers, as well as several other diseases (30–37). However, in 2016, a liver cancer trial for a miR-34a-5p mimic reported serious immune system-related adverse events and the death of four patients (101). Since 2016, researchers have investigated the causes of the adverse events in this trial, and new approaches are being explored to deliver miRNA-based therapies that would limit dangerous side effects in the future. In the cancer context, this includes encapsulating miRNA mimics in nanoparticles that are designed to delay release of the miRNA until they reach the tumor environment (102). Research is also ongoing to identify miRNA sequence modifications and new encapsulation methods that would enhance cellular uptake and effectiveness of miRNA mimics inside of cells: curtailing degradation by nucleases in the systemic circulation and degradation within the endosomal compartment (103).

To target microbial infections in the lung or gut, delivery of miRNA mimics faces the additional challenge of permeating mucus to reach colonizing microbes. The challenge is especially acute in a disease like CF where mucus is hyper-viscous. Nanoparticles engineered to breach the mucus barrier might overcome this problem (104–106). Also, to limit the likelihood of off-target side effects, delivery vehicles should be designed to target bacterial cells specifically (107), just as miRNA-based cancer therapeutics are being designed for selective release in the tumor environment.

The Rocket-miR application adds to our suite of existing community-oriented bioinformatic tools—including three other R shiny applications (ScanGEO, GAPE, CF-Seq)

that make insights from public transcriptomic data on human pathogens more accessible to non-computational scientists, and a fourth (ESKAPE Act Plus) that enables researchers without computational experience to perform pathway activation analysis for ESKAPE pathogens (108–111). We envision Rocket-miR and these other applications as ongoing projects—tools that researchers studying any microbial pathogen can use and extend, to the ultimate benefit of patients with drug-resistant infections. Currently, the Rocket-miR application includes target prediction data for 24 CF pathogens and the so-called ESKAPE pathogens, renowned for their antibiotic resistance and tendency to cause hospital-acquired infections (5, 6). For other human pathogens of interest, researchers can follow the methods outlined in this paper, and the instructions in the Github repository, to build their own version of the application: https://github.com/samlo777/Rocket-miR

## MATERIALS AND METHODS

### Acquiring public sequence data

The Rocket-miR application includes target prediction data for 24 human pathogens. Making these target predictions required downloading public data from the National Center for Biotechnology Information (NCBI) Assembly database. For each species and particular strain of interest, the genomic coding sequence (CDS) file was downloaded. The coding sequence is the string of nucleotides in an mRNA transcript that corresponds directly to the string of amino acids in the associated protein. Thus, it reflects the state of the transcript after non-coding sequences have been removed by RNA processing (112, 113).

Although researchers have traditionally operated on the assumption that human miRNAs target the 3′ untranslated region of human transcripts, more recent research suggests that miRNAs can target other elements of the transcript such as the coding region (114, 115). Furthermore, IntaRNA target prediction data from our prior study on let-7b-5p targeting of *P. aeruginosa* suggest that in *P. aeruginosa*, human miRNAs are far more likely to interact with the coding region than the untranslated region of the transcript (38). Thus, the prediction data in the Rocket-miR application are based on coding sequences in CF pathogens (though future efforts to repurpose the application could assess the predicted targeting of non-coding regions as well).

The CDS file contains the nucleotide sequence of all known protein-encoding genes for a given pathogen strain. It also contains gene identifier metadata that enables the genes to be mapped to KEGG biological pathways. For each of the species in the application, we chose strains that have annotations in the KEGG database, enabling us to make predictions about how specific miRNAs target KEGG pathways.

The human miRNA sequence data were downloaded from the database MiRGeneDB, which recognizes 630 different human miRNAs as mature miRNA sequences. All the human miRNAs in MiRGeneDB were manually curated by the creators of the database. The creators of MiRGeneDB have noted that other miRNA databases, which encourage community submission of miRNA sequences and include several thousand putative human miRNAs, contain many false positives. Certain sequences labeled as miRNAs may belong to other classes of small RNAs. For this reason, we have chosen the 630 miRNAs in MiRGeneDB as the gold standard for human miRNA sequences (116). That said, researchers may be interested in repurposing the application to predict targets of other small miRNAs. Researchers may also design new miRNA sequences (miRNA mimics) in an attempt to improve on the natural targeting capacity of human miRNAs. In both cases, IntaRNA can be used to make target predictions, which can then be visualized with the Rocket-miR application.

## Predicting miRNA targets

The IntaRNA target prediction algorithm, designed by the Backofen Lab at the University of Freiburg, predicts hybridization between two RNAs, including miRNA and mRNA (58, 117). It considers not only the propensity for nucleotides in the two interacting RNAs to associate, but also the secondary structure of each individual RNA. Before two RNAs can interact, each must adopt an "open," linearized conformation so that nucleotides in the two molecules are exposed to one another for interaction. If a given RNA has a higher tendency to fold up on itself and form interactions between its own nucleotides, it is less likely to interact with other RNAs, and therefore its predicted interaction score is less favorable.

IntaRNA was selected as a target prediction algorithm for several reasons. First, databases that provide ready prediction data to the public are limited in scope. The database miRDB, for example, presents millions of predicted targets of miRNAs across five species (human, mouse, rat, dog, and chicken) (92, 93). However, databases like miRDB are limited to model organisms and other popular species—not microbial pathogens such as those that infect people with CF and other diseases. Though other prediction algorithms like IntaRNA exist that can predict interactions between different RNAs for any species, IntaRNA was chosen because it considers both the nucleotide sequence of pairing miRNAs and the intramolecular interactions of those miRNAs as outlined in the previous paragraph. Other available algorithms—RNAhybrid and RNAplex, for example—tend to focus on nucleotide pairing alone (118, 119). Finally, the use of IntaRNA is supported by our own prior experience. We have used the algorithm in prior studies to predict miRNA targeting, and these predictions were born out by the results of wet bench experiments (38, 52). This gives us confidence in the algorithm's predictive capacity for human pathogens.

For each RNA-RNA interaction, IntaRNA calculates an energy score. This energy score is more negative when the interaction between the miRNA and mRNA is stronger. For example, an interaction between a miRNA and mRNA that has an energy score of $-15$ is predicted to be a stronger interaction than a miRNA-mRNA pair with an energy score of $-8$. Only negative energy scores are possible. Although RNA molecules possess the nucleotide uracil (U) instead of thymine (T), some RNA sequences in public databases (including those featured in the application) contain Ts in the place of Us. The IntaRNA algorithm recognizes this possibility and automatically converts all thymine nucleotides to uracil in order to predict RNA interactions (see note in the IntaRNA Github Repository: https://github.com/BackofenLab/IntaRNA).

The energy cutoffs (strong: $-16.98$, medium: $-15.35$, weak, $-13.47$) in the application are set based on data from our prior work, in which we determined that *Pseudomonas aeruginosa* cells over-expressing let-7b-5p significantly downregulated a set of 48 proteins, with the targeted proteins having a distribution of predicted IntaRNA energy scores [Appendix, pg. 18 (38, 38)]. The most stringent cutoff in the application corresponds to the first quartile value in this distribution, the medium targeting cutoff to the median, and the least stringent cutoff to the third quartile value. It is important to note that we chose to limit our analysis to proteins with predicted energy scores less than the least stringent cutoff to be downregulated significantly. Using more stringent cutoffs would increase confidence in the likelihood of targeting but identify fewer candidate miRNAS, a tradeoff that might be suboptimal for species that are less effectively targeted by human miRNAs.

The IntaRNA program is typically run in a terminal window. When calling the program, a set of query sequences and a set of target sequences are specified in a text-based file. In the context of this study, the query sequence was always set up to contain the set of 630 human miRNA sequences, and the target file contained coding sequences from the different microbial pathogens included in the Rocket-miR application. This computational task was performed on a high-performance computing server. To make the computation more feasible (screening all 630 human miRNAs against 1,000 of mRNA transcripts at once would take up an excessive amount of memory on the server),

each CDS file was divided into many small files of 250 mRNA transcripts a piece. Then, the 630 miRNAs were screened against each set of 250 transcripts individually—each with a single call of the IntaRNA algorithm. Using this approach, a microbial pathogen with 4,000 genes requires the IntaRNA algorithm to be applied 16 separate times. Ultimately, the 250-transcript files were stitched back together in a single "Energy Table" for inclusion in the application. All codes used in the command line to run IntaRNA on pathogens of interest, as well as the R script used to consolidate the 250-transcript files into Energy Tables, and data files including the energy tables themselves are available in our GitHub repository: https://github.com/samlo777/Rocket-miR

## Gathering gene and pathway annotations

As mentioned above, the microbial strains featured in the application were chosen because they had associated KEGG pathway annotations. This enables users to visualize pathway targeting for their pathogens of interest in addition to the targeting of individual genes. Once the target prediction data were generated for a particular pathogen using IntaRNA, and the "Energy Table" was consolidated, the gene IDs were extracted from that energy table using regex functions from the "stringr" package in R (120). There are multiple gene IDs housed in the CDS files, though the kinds of IDs present can differ between species. When available, the NCBI Gene ID was extracted. If not, the RefSeq protein ID was extracted.

Using the ID Mapping feature on the Uniprot website (https://www.uniprot.org/id-mapping), NCBI Gene IDs or RefSeq protein IDs were mapped first to Uniprot IDs, and then to KEGG gene IDs in a multistep process (121). This ID mapping was performed for all the species. In addition, the R "EnrichmentBrowser" package was used to identify the genes on each KEGG pathway (each pathway has its own pathway identifier) (122). Ultimately, the KEGG gene IDs and KEGG pathway identifiers were all merged into a single table. All gene annotation files are available in the Github repository.

## Application development

The Rocket-miR application was developed in multiple stages. The first, outlined in the sections above, involved gathering the target prediction and annotation data together in the right format. Once the data were prepared for several pathogens, the various sections of the application were prototyped. The application was carefully designed so that when the time came for new prediction data to be added, it could be seamlessly integrated without changing any of the application code.

The Rocket-miR application, like all R shiny applications, requires a specific code architecture—a single file labeled app.R with a section for user interface (UI) code (defining the appearance of the application) and a subsequent section for server code (defining the functional code of the application). In addition to the app.R file, a "Data Processing" script was prepared to generate the energy tables for inclusion in the application, and a "Data Setup" script to combine the energy tables with gene annotations. The "Data Setup" file also performed some preliminary calculations on the energy table data so that users of the application could run the application and visualize data more quickly instead of waiting for those calculations to run. All these code files are available in the Github repository.

The application, developed in R using the RStudio IDE, employs a number of publicly available, open-source R packages (123, 124). In addition to the "shiny" package, which is required for all shiny applications, the Rocket-miR application makes use of the "shinydashboard" package for its application layout (125, 126). The "shinyjs" package was used to enable users to switch between different modules in the application (127). The "shinyBS" package was used to create custom buttons (128). The "DT" package was used to generate data tables that are filterable and searchable (129). The "stringr" package was used to simplify the text in these tables (e.g., from long gene annotations to a simple gene name) (120). The "shinycssloaders" package enabled loading animations (130). Finally, for data analysis and visualization, the "plotly" package was used to create

interactive figures, and the "kmer" package was utilized to analyze the structure of miRNA sequences (131, 132).

## *In vitro* validation experiment

PA14 was electroporated to introduce a pMQ70 expression vector (with arabinose-inducible promoter and carbenicillin resistance as selectable marker) containing either the mature let-7b-5p sequence as an insert (PA14-let-7b-5p) or no insert (PA14-Empty vector), as described previously (38). The glycerol stocks of these two strains were first streaked out on two different Luria broth (LB) agar plates containing 300 µg/mL carbenicillin. The colonies from the plates were then inoculated in LB broth containing 150 µg/mL carbenicillin for two overnight grow-ups. Twenty-five thousand CFU per mL of the control strain or PA14-let-7b-5p was then plated in MBEC Assay Biofilm Inoculator Plates (Innovotech) with 100 µM arabinose to induce the expression from the promoters and with or without 1.2 µg/mL aztreonam. The plate was placed on a shaker at 110 rpm and 37°C. Twenty-four hours later, the coverlid, composed of the pegs, was washed twice with PBS++ (ThermoFisher DPBS with added calcium and magnesium). The coverlid was then placed into a new 96-well plate containing 200 µL fresh LB/well and sonicated for 10 min in a water-bath sonicator (VWR Symphony model). The LB from the wells was serially diluted, plated on LB agar plates, and incubated overnight at 37°C. CFU were counted and final counts were adjusted by multiplying the dilution factor. Data were analyzed using GraphPad Prism (version 9.1.2; San Diego, CA) and the R software environment for statistical computing and graphics version 4.1.0 (123). The R package MASS was used to calculate $P$-values from the binomial generalized linear models (133).

## ACKNOWLEDGMENTS

Support for these studies was provided by the Cystic Fibrosis Foundation (STANTO19R0–CFF), the NIH (grants P30 DK117469, R01 HL151385, R35 GM142685), and The Flatley Foundation.

S.L.N. developed the Rocket-miR application and wrote the manuscript apart from the case studies. T.H.H. and K.K. helped guide the initial formulation of the project. T.H.H. provided helpful feedback on the application and manuscript throughout their development. K.K. contributed code enabling the miRNA target predictions that form the basis of the application. S.S. contributed the first two case studies, while C.J.L. and B.D.R. contributed the third. T.H.H., K.K., S.S., C.J.L., B.D.R., and B.A.S. all provided input on the development of the application and contributed to manuscript preparation for submission.

## AUTHOR AFFILIATION

[1]Department of Microbiology and Immunology, Geisel School of Medicine, Dartmouth College, Hanover, New Hampshire, USA

## AUTHOR ORCIDs

Samuel L. Neff  http://orcid.org/0000-0002-5993-8445
Thomas H. Hampton  http://orcid.org/0000-0003-0543-402X
Katja Koeppen  http://orcid.org/0000-0003-1094-0621
Bruce A. Stanton  http://orcid.org/0000-0002-1661-407X

## FUNDING

| Funder | Grant(s) | Author(s) |
|--------|----------|-----------|
| Cystic Fibrosis Foundation (CFF) | STANTO19R0 | Samuel L. Neff |
| | | Thomas H. Hampton |
| | | Katja Koeppen |

| Funder | Grant(s) | Author(s) |
|---|---|---|
| | | Sharanya Sarkar |
| | | Bruce A. Stanton |
| HHS \| National Institutes of Health (NIH) | P30 DK117469, R01 HL151385 | Thomas H. Hampton |
| | | Katja Koeppen |
| | | Sharanya Sarkar |
| | | Bruce A. Stanton |
| | | Samuel L. Neff |
| HHS \| National Institutes of Health (NIH) | R35 GM142685 | Casey J. Latario |
| | | Benjamin Ross |
| The Flatley Foundation | | Samuel L. Neff |
| | | Thomas H. Hampton |
| | | Katja Koeppen |
| | | Sharanya Sarkar |
| | | Bruce A. Stanton |

## AUTHOR CONTRIBUTIONS

Samuel L. Neff, Conceptualization, Data curation, Formal analysis, Methodology, Project administration, Software, Supervision, Visualization, Writing – original draft, Writing – review and editing | Thomas H. Hampton, Conceptualization, Methodology, Supervision, Writing – review and editing | Katja Koeppen, Conceptualization, Methodology, Writing – review and editing | Sharanya Sarkar, Formal analysis, Validation, Visualization, Writing – original draft, Writing – review and editing | Casey J. Latario, Formal analysis, Validation, Visualization, Writing – original draft, Writing – review and editing | Benjamin D. Ross, Funding acquisition, Supervision, Validation, Writing – review and editing | Bruce A. Stanton, Funding acquisition, Project administration, Supervision, Validation, Writing – review and editing

## DATA AVAILABILITY

The source code for the Rocket-miR application was written in R, using the R Studio integrated development environment. The application is accessible online at the following address: http://scangeo.dartmouth.edu/RocketmiR/. The source code and application data are accessible in our Github Repository and can be run independently using RStudio on a personal device by following the instructions in the repository's README file: https://github.com/samlo777/Rocket-miR.

## ADDITIONAL FILES

The following material is available online.

### Supplemental Material

**Supplemental figures (mSystems00653-23-s0001.pdf).** Fig. S1 to S5.

### Open Peer Review

**PEER REVIEW HISTORY An accounting of the reviewer comments and feedback. (review-history.pdf).**

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
