## [Reviewer comments · mSystems]

Rocket-miR, a Translational Launchpad for miRNA-based Antimicrobial Drug Development

Samuel Neff, Thomas Hampton, Katja Koeppen, Sharanya Sarkar, Casey Latario, Benjamin Ross, and Bruce Stanton

Corresponding Author(s): Bruce Stanton, Geisel School of Medicine at Dartmouth

Review Timeline:

Submission Date:	June 22, 2023
Editorial Decision:	September 20, 2023
Revision Received:	October 6, 2023
Accepted:	October 6, 2023

Editor: Jack Gilbert

Reviewer(s): The reviewers have opted to remain anonymous.

Transaction Report:

DOI: <https://doi.org/10.1128/msystems.00653-23>

September 20, 2023

Prof. Bruce A. Stanton
Geisel School of Medicine at Dartmouth
Hanover

Re: mSystems00653-23 (Rocket-miR, a Translational Launchpad for miRNA-based Antimicrobial Drug Development)

Dear Prof. Stanton:

Firstly I am so sorry this took so long to make a decision on. The first editor tried to find people to review and after inviting 16 people who all declined eventually decided that they couldn't make a decision without input. I asked one more editor to decline. Luckily, I am more confident in my own opinion and after reading the paper I am confident it can be published. My only concern (the paper is very well written and presented) is that no attempt appears to have been made to compare the results of in vitro experiments to the predictions. As you must have in vitro data on the activity of let-7b-5p against *Pseudomonas*, is it not possible to compare the results against the prediction? If this is in the paper I missed it. Hence I am sending back for minor edits to ensure that there are no changes you would like to make prior to acceptance, and to be sure you have the opportunity to address my concern.

Sincerely

Jack Gilbert

Thank you for submitting your manuscript to mSystems. We have completed our review and I am pleased to inform you that, in principle, we expect to accept it for publication in mSystems. However, acceptance will not be final until you have adequately addressed the reviewer comments.

Preparing Revision Guidelines

Please return the manuscript within 60 days; if you cannot complete the modification within this time period, please contact me. If you do not wish to modify the manuscript and prefer to submit it to another journal, please notify me of your decision immediately so that the manuscript may be formally withdrawn from consideration by mSystems.

Sincerely,

Jack Gilbert

Editor, mSystems

Journals Department
Reviewer comments:

Dear Dr. Gilbert,

We appreciate your feedback on the Rocket-miR manuscript and have addressed your suggestion about incorporating in vitro data to test Rocket-miR predictions.

The updated version of the manuscript contains an additional section titled 'Experimental Validation of Rocket-miR Predictions' with in vitro data on the efficacy of human miRNA let-7b-5p against cystic fibrosis (CF) pathogen *P. aeruginosa*. The effects of let-7b-5p were tested on both planktonic and biofilm growth, and in the presence or absence of aztreonam, a beta-lactam antibiotic commonly prescribed for people with CF.

Let-7b-5p reduced planktonic growth on its own but had a heightened effect in the presence of aztreonam. Let-7b-5p also inhibited biofilm formation in the presence of aztreonam. These results are in line with Rocket-mir predictions that let-7b-5p targets the beta lactam resistance pathway in *P. aeruginosa*.

In addition to the new results, we also included an additional section of the methods to outline the experiments in detail.

Please find attached the latest manuscript version and figures – and thank you again for your review of the paper and helpful suggestions.

Sincerely,

Samuel Neff
Lewis-Sigler Institute of Integrative Genomics, Princeton University
sam.neff@princeton.edu

Corresponding Author:
Bruce Stanton, Ph.D.
Geisel School of Medicine, Dartmouth College
Bruce.A.Stanton@dartmouth.edu

October 6, 2023

Prof. Bruce A. Stanton
Geisel School of Medicine at Dartmouth
Hanover

Re: mSystems00653-23R1 (Rocket-miR, a Translational Launchpad for miRNA-based Antimicrobial Drug Development)

Dear Prof. Bruce A. Stanton:

Your manuscript has been accepted, and I am forwarding it to the ASM Journals Department for publication. For your reference, ASM Journals' address is given below. Before it can be scheduled for publication, your manuscript will be checked by the mSystems production staff to make sure that all elements meet the technical requirements for publication. They will contact you if anything needs to be revised before copyediting and production can begin. Otherwise, you will be notified when your proofs are ready to be viewed.

If you would like to submit a potential Featured Image, please email a file and a short legend to mSystems@asmusa.org. Please note that we can only consider images that (i) the authors created or own and (ii) have not been previously published. By submitting, you agree that the image can be used under the same terms as the published article. File requirements: square dimensions (4" x 4"), 300 dpi resolution, RGB colorspace, TIF file format.

We recognize that the video files can become quite large, and so to avoid quality loss ASM suggests sending the video file via <https://www.wetransfer.com/>. When you have a final version of the video and the still ready to share, please send it to mSystems staff at mSystems@asmusa.org.

Sincerely,

Jack Gilbert
Editor, mSystems

Journals Department
E-mail: mSystems@asmusa.org